# Varieties of Lettuce Forming Distinct Microbial Communities Inhabiting Roots and Rhizospheres with Various Responses to Osmotic Stress

Jana Žiarovská [1], Lucia Urbanová [2], Dagmar Moravčíková [1], Renata Artimová [2], Radoslav Omelka [3] and Juraj Medo [4],*

1   Institute of Plant and Environmental Sciences, Faculty of Agrobiology and Food Resources, Slovak University of Agriculture in Nitra, Tr. A. Hlinku 2, 94976 Nitra, Slovakia
2   Research Centre Agrobiotech, Slovak University of Agriculture in Nitra, Tr. A. Hlinku 2, 94976 Nitra, Slovakia
3   Department of Botany and Genetics, Faculty of Natural Sciences and Informatics, Constantine the Philosopher University in Nitra, 94974 Nitra, Slovakia
4   Institute of Biotechnology, Faculty of Biotechnology and Food Sciences, Slovak University of Agriculture in Nitra, Tr. A. Hlinku 2, 94976 Nitra, Slovakia
*   Correspondence: juraj.medo@uniag.sk

**Abstract:** A plant microbiome is an important factor in plant growth, stress resistance, health status, and consumer quality and safety. The rhizosphere microbiome evolves in a negotiation between microbial communities that inhabit soil and plant root tissue. In this study, the rhizosphere and root internal tissue microbiome of six varieties of lettuce were analyzed in normal conditions and under salinity stress. The metabarcoding analysis used 16S rRNA gene and ITS2 region sequencing. The microbiomes of root samples were significantly less diverse with different members of the community compared to those of the rhizosphere. A significant effect of lettuce variety was found on the diversity index for bacteria and fungi. Varieties formed very different communities of bacteria in roots. *Pseudomonas*, *Herbaspirillum*, *Mycobacterium*, potentially pathogenic *Enterobacter*, and other genera were more prevalent in certain varieties. Salinity stress had a significant negative impact on bacterial diversity and community composition, whereas the diversity of fungi has not changed significantly, and the fungal community has changed less than the bacterial one. Changes were more evident in varieties that were more resistant to salinity stress than in sensitive varieties.

**Keywords:** *Lactuca sativa* L.; 16S rRNA sequencing; ITS region sequencing; endophytes; rhizosphere; osmotic stress

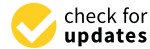



## 1. Introduction

Lettuce (*Lactuca sativa* L.) is one of the most freshly consumed vegetables [1]. It is an important source of fiber, vitamins, antioxidants, and other nutrients that positively impact human health and well-being [2,3]. Lettuce comprises many varieties with diverse phenotypes [4]. Globally, lettuces are grown in an area of about 1.8 million hectares in field or greenhouse conditions [5]. They are grown in many soil or substrate types, including degraded saline soil fields.

Plants and the environment are inhabited by various microbial species that form complex microbial communities called microbiomes. Plant microbiomes are currently understood as the sum of all microbial genomes that inhabit plant leaves, fruits, and rhizospheres in interconnected microbial network complexes [6–9]. Members of these communities affect plants during their growth and can act as neutral, beneficial, or harmful. Plant-associated microorganisms have been proven to affect the entire plant's growth process, nutrition, productivity, and disease resistance [10,11].

Research of plant-associated microbiomes is currently focused mainly on the rhizosphere [12]. The rhizosphere is an interaction zone between the soil and the plants, rich

in microbial diversity. The decomposition of organic matter, nutrient cycling and other ecologically important microbe-driven processes are very intensive here. Available data suggest that plants are involved in the selection process when the microbiome of the rhizosphere is distinguished from soil. Plant products such as carbohydrates, amino acids or other secondary root-secreted metabolites (rhizodeposits) create specific conditions for microbial growth [13]. It is assumed that plants could modulate the rhizosphere microbiota by selective stimulation of microbiome members beneficial for their health and growth [14]. Rhizodeposition leads to development of specific rhizosphere microbiomes, and it is also the first step in selection of endophytic microbial assemblage within root tissues. Some pilot studies showed that the plant genotype is responsible for the second step of root microbiome shaping [15]. The effect of plant genotypes on root-associated microbiomes was found primarily on longer living plants such as trees and perennial crops [16,17] but also in some annual crops [18]. It is mainly connected to the different physiology of plant genotypes [19].

Plant microbiomes not only affect the plant itself, but some microbiome members can directly or indirectly affect human health [14]. Plant tissues may be successfully colonized by pathogenic bacteria such as *Salmonella*, *Escherichia coli*, *Campylobacter*, *Listeria*, or *Enterobacteriaceae* [20,21]. Members of other bacterial genera (*Pseudomonas*, *Burkholderia*, *Pantoea* and others) may act as potential pathogens, although they are common members of the rhizosphere and plant microbiomes [22–24]. Even though some contamination may occur during handling [25], it is usually associated with agricultural practices such as organic fertilizers [26] or untreated irrigation water [27]. As lettuce is consumed almost solely in its fresh state without any heat treatment, any contamination by human pathogenic bacteria may be dangerous [28]. The rhizosphere is considered to be a playground in which complex microbial interactions may allow or suppress the development of plant or human pathogens [29].

Salinity of soil is one of the key factors that limits agricultural use of the soil in certain areas [30]. Salinity stress affects plant growth, productivity, and health [31]. Selection of saline-tolerant varieties can open a way to use even degraded saline soil. Some lettuce varieties have been already selected to tolerate high salinity, and future research is desirable [32]. Recent studies showed distinctive composition of microbiomes in saline soils [33]. It is also associated with the specific development of plant microbiomes under salinity stress [34,35]. Supposedly, such microbiomes help the plant to withstand the harmful conditions of saline soil [36]. Plants participate in this selection primarily through their products such as exopolysaccharides, playing a role in root biofilm formation [37]. Research of microbiomes in saline-stressed plants can be useful in future selections of microbiome members that can increase plant tolerance to salinity stress [38,39]. Several strains of plant growth-promoting rhizobacteria with such ability were already described [40,41].

Understanding the biological processes that shape the structure and dynamics of a microbiome of the rhizosphere is a fundamental step to ensure plant productivity and in producing safe food [10]. Although some studies examined the interaction between plant genotype and salinity, differences in tolerance were not reported for used genotypes. Salinity-tolerant varieties should maintain the development of microbiomes in salt soils due to the ability to support bacteria with root exudates.

The aim of the study was to characterize and compare the development of microbiomes in the rhizosphere and internal root tissue of six different varieties of lettuce and to assess the reaction of these microbiomes to salinity stress. We hypothesize that the lettuce microbiome depends on the variety, and the reaction to salinity is different in varieties that are naturally tolerant of high salinity.

## 2. Materials and Methods

Six varieties of lettuce—Bibb, SM09PA, Romana Larga Blanca, Dark Green Romaine, Pavane, and Sentry were used in the research. The first three have a higher salt tolerance according to Xu and Mou [32] and Adhikari et al. [42]. The lettuce seeds were surface

sterilized for 2 min in 0.5% sodium hypochlorite and then washed three times with sterile distilled water. Seeds were pre-germinated on filter paper in a Petri dish moistened by sterile water. Pre-germination took seven days in a climate chamber under a 16 h light and 8 h dark regime, at 20 and 10 °C, respectively. The germinated seeds were then transplanted into $90 \times 80 \times 80$ mm pots (single seed per pot) filled with commercially available growing substrate (gardening substrate with vitality complex, AgroCS, Říkov, Czech Republic).

Lettuce was grown in six pots for each variety. Growing conditions were based on the modified protocol from Wei et al. [43], with light cycles of a 16 hour's day and 8 hour's night, a temperature of 18–20 °C, and 60% humidity. The air was purified from pathogens. The plants were watered every two days with sterile distilled water to maintain the desired humidity. After two weeks, watering in an experimental group (three pots per variety) with a solution of 100 mM NaCl induced osmotic stress whereas the control group (3 pots) was watered with distilled water. According to the protocol [43], samples were watered on the 1st, 4th, and 7th day of the stress period by 30 mL and then by 40 mL on days 10 and 13. Samples were collected two weeks after the stress induction. The total conductivity of $EC_{1:5}$ measured in soil water extract (ratio 1:5 $w/v$) reached $4.64 \pm 0.31$ dS/m in the treated samples compared to $1.36 \pm 0.16$ dS/m in the control soil after the stress period.

Each variety provided 2 types of samples—the internal root microbiome and the root surface soil microbiome (the rhizosphere). Roots were cleaned from the soil softly by a brush, and rhizosphere samples were prepared as the leachate of the root surface into 0.9% saline solution in 50 mL falcon tubes. The suspension of soil particles containing the microbiome of the rhizosphere was centrifuged for 20 min at $6000 \times g$, and the resulting pellets were used for DNA extraction. The roots were then rinsed several times with sterile distilled water, the surface was sterilized with 2% sodium hypochlorite solution for 5 min, and was again rinsed with water several times. The prepared roots were used for DNA extraction.

*2.1. DNA Extraction, Amplification, and Sequencing*

DNA was extracted using the MOBIO Powersoil DNA extraction kit (Qiagen, Hilden, Germany). Garnet particles in the kit were replaced by 2 mm diameter zirconium oxide beads to better disintegrate the root tissue. In total, 250 mg of sample (root tissue or rhizosphere pellet) was homogenized with the BeadBug homogenizer (Benchmark scientific, Sayreville, NJ, USA).

General bacterial primers 515F and 806R [44] enhanced by 8 bp identification sequence (tag) were used for amplification of V4 region of 16S rRNA gene. For analysis of fungal community, primers g ITS7 and ITS4 [45] were used in the amplification of the ITS2 region (Supplement Table S1). The composition of the PCR mixture was as follows: 15 µL KAPA HIFI HS MIX 2X (Roche, Indianapolis, IN, USA), 4 µL of each primer with a concentration of 2.5 µM, and 1 µL of extracted DNA. The amplification was carried out in the Stratagene mx3005p thermal cycler (Agilent, Santa Clara, CA, USA) with the following configuration. Initial denaturation for 90 s at 98 °C was followed by 35 cycles of denaturation for 15 s at 98 °C, annealing for 15 s at 62 °C, and by extension for 15 s at 72 °C. The final extension was 2 min at 72 °C. The PCR products were purified using a PCR purification kit (Jena Bioscience, Jena, Germany). Then, the PCR products were quantified by qubit (Thermo scientifics, Waltham, MA, USA), diluted to the same concentration and pooled together. Illumina adapters were attached by TruSeq LT PCR free kit (Illumina, San Diego, CA, USA) with modification involving the skip of DNA fragmentation and size selection. The library was quantified by qPCR using NebNext Quantification kit (New England BioLabs, Ipswich, MA, USA), diluted to 4 nM concentration, and denatured. The MiSeq Reagent Kit v3 (600-cycle) was used for sequencing and a 20 pM library with 1% PhiX spike was loaded into the cartridge.

## 2.2. Sequences Analysis

Acquired data was processed in the SEED2 environment (version 2.12) [46]. Forward and reverse readings were joined with join2fastqc and sequences with overall quality less than Q30 were removed from further analysis. The primers were removed, and the sequences were processed by the Vsearch [47] algorithm to detect chimeras, which were also removed from further analysis. Chimera-free sequences were clustered to operational taxonomic units (OTUs) using Vsearch set at 97% similarity level. The most abundant sequence was found in each cluster (OTU), and such sequences for each OTU were identified using the RDP classifier [48]. Sequences of chloroplasts, mitochondria, and the ITS region of plants were removed from further analysis. The most abundant sequences in each OTU were aligned with MAAFT [49], and a phylogenetic tree was constructed using PhyML [50]. Using the phylogenetic tree and OTU table, a weighted Unifrac [51] distance matrix was calculated in the R statistical environment [52]. Non-metric multidimensional scaling (NMDS) analysis and permutational multivariate analysis of variance (PERMANOVA) statistics based on the Unifrac matrix were obtained with the package Vegan [53]. Heatmaps were made using the Heatmap3 package [54] in R. Linear discriminant analysis effect size (LefSe) [55] was used to compare the abundance of taxa between varieties and the discovery of biomarkers. EdgeR [56] was used to analyze changes in tax abundance due to the salinity stress in each variety.

Tables of OTUs were rarefied to the lowest sequence count among samples for alpha diversity assessment. Alpha diversity was described by OTU Richness, Shannon's index, and Pielou's corrected evenness. The indices were statistically evaluated using the multifactor ANOVA in R.

## 3. Results

There were 1,104,413 and 1,137,914 high-quality chimera-free sequences acquired for the ITS2 region and the 16S rRNA gene, respectively. All sequence data was submitted to GenBank databases as part of BioProject PRJNA893639. The mean number of sequences per sample was 15,339 for ITS2 and 15,804 for the 16S rRNA gene. However, samples taken from the plants' roots contained 12–30% of sequences identified as lettuce ITS2, which were removed before analysis. Similarly, 16S rRNA gene sequences from root samples showed a considerably high portion of chloroplast (21–29%) and mitochondria (16–23%) sequences that needed to be removed prior to the analysis. Rhizosphere samples contained less than 10% of such sequences. In total, 4429 OTUs were generated from 16 rRNA gene sequences, and 1830 OTUs were found for ITS sequences.

## 3.1. Diversity and Structure of Bacterial Community

Using 3-factor ANOVA, the sample type was identified as the most important factor affecting diversity indices ($p < 0.001$) (Table 1). Furthermore, the effect of osmotic stress was highly significant ($p < 0.001$). Variety did not affect the richness significantly ($p = 0.183$) but evenness and Shannon's index varied significantly ($p < 0.001$) among lettuce varieties. Moreover, highly significant interactions between variety and sample type, variety and stress, as well as between sample type and stress (all $p < 0.001$) were detected. These interactions indicate a mixed response of microbiomes to salinity stress among lettuce varieties. Moreover, this response of microbiomes in the rhizosphere zone and root tissue is clearly not the same. The lowest values of indices were found in the root samples of salinity tolerant varieties in saline conditions (Supplement Table S2).

**Table 1.** Alpha diversity indices of bacterial microbiomes in root and rhizosphere of six varieties of lettuce under normal conditions and salinity stress.

| Factor | Variant | Richness | Shannon's Index | Evenness |
|--------|---------|----------|-----------------|----------|
| Sample type | Root | 114 a * | 7.83 a | 0.721 a |
| | Rhizosphere | 158 b | 8.42 b | 0.786 b |
| | | $p < 0.001$ | $p < 0.001$ | $p < 0.001$ |
| Variety | Bibb | 486 a | 8.15 ab | 0.762 ab |
| | Dark Green Romaine | 521 a | 8.44 b | 0.805 b |
| | Romana Larga Blanca | 490 a | 8.07 ab | 0.757 ab |
| | Pavane | 496 a | 7.95 a | 0.722 a |
| | Sentry | 518 a | 8.11 ab | 0.736 a |
| | SM09 PA | 496 a | 8.02 a | 0.738 a |
| | | $p = 0.183$ | $p = 0.006$ | $p < 0.001$ |
| Salinity stress | No | 529 b | 8.29 b | 0.769 a |
| | Yes | 474 a | 7.95 a | 0.738 a |
| | | $p < 0.001$ | $p < 0.001$ | $p = 0.002$ |

* Averages followed by the same letter are not significantly different from each factor (ANOVA, Tukey test $\alpha = 0.05$)

The primary driver of the differences in the bacterial community composition was the variety (PERMANOVA $p < 0.001$; $R^2 = 0.15$). In addition, the sample type and salinity stress significantly affected the community, and all interactions were significant. Multivariate dispersion of root samples was significantly wider than dispersion of rhizosphere samples (BETADISPER $p < 0.001$). The difference in dispersion can affect PERMANOVA results, so root tissue and rhizosphere samples were analyzed separately (Figure 1). Variety was the most significant factor in both sample types, and differences were confirmed by pairwise comparison (Supplement Table S3). For certain varieties (Bibb, Romana Larga Blanca, Dark Green Romaine, and SM09PA), salinity stressed samples of lettuce roots were clearly separated from control root samples according to PERMANOVA results ($p < 0.001$). The clustering of the varieties was also observed in the rhizosphere samples. The effect of salinity stress was not clear, despite the significant interaction between stress and variety ($p = 0.007$).

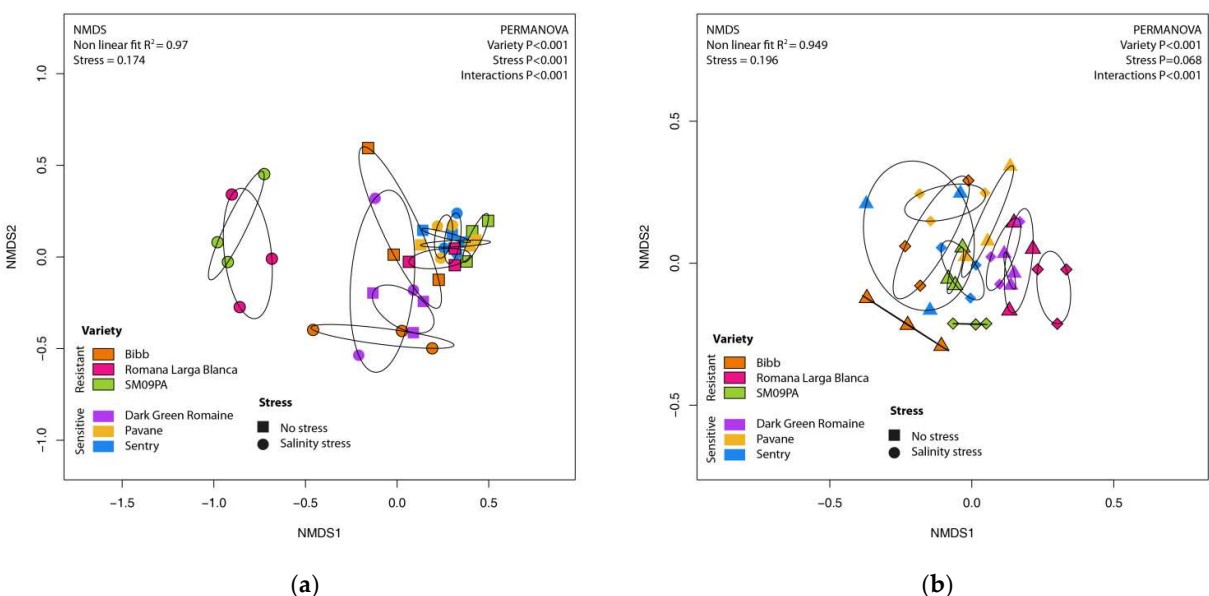

(**a**)          (**b**)

**Figure 1.** NMDS scatterplot of bacterial community in roots (**a**) and rhizosphere (**b**) of six lettuce varieties in normal conditions and under salinity stress.

Rhizosphere microbiome samples contained slightly less *Proteobacteria* and *Firmicutes* but more *Actinobacteria*, *Chloroflexi*, and *Planctomyces* phyla (Figure 2). Analysis at the genera level led to clear clustering depending on the sample type and variety (Figure 3). The most common genera were *Acidobacteria* group GP6, *Pseudomonas*, *Herbaspirillum*, and *Flavobacterium*. In the rhizosphere of non-stressed plants, LefSe analysis showed *Mycobacterium* and *Sedimenticola* to be biomarkers for the Pavane variety, and *Herbaspirillum* and *Conexibacter* for the Sentry variety. Romana Larga Blanca's variety biomarkers were *Ktenobacter* and *Streptomyces*. *Limnobacter's* and *Aquabacterium's* were selected for the Bibb variety. *Dokdonella* and *Salinibacterium* were biomarkers for the Dark Green Romaine variety, and *Blastocladella* for the SM09PA variety. It was not fully consistent with the analysis of non-stressed plant root samples where *Pseudomonas*, *Mycobacterium*, and *Actinomadura* have been found as biomarkers for the Pavane variety. *Conexibacter*, *Burkholderia*, *Sacharibacteria*, *Halomonas*, and others were biomarkers for the Sentry variety whereas *Ktedonobacter*, *Otitutus*, and *Aridibacter* were biomarkers for the Dark Green Romaine variety. *Ornatilinea* and *Thermogutta* were selected for the Bibb variety, and *Cellvibrio*, *Staphylococcus*, and *Methylophilus* for the SM09PA variety. Bar charts (Figure 4) show the first 20 biomarker OTUs for varieties in rhizospheres and root samples, which reflect the biomarker genera.

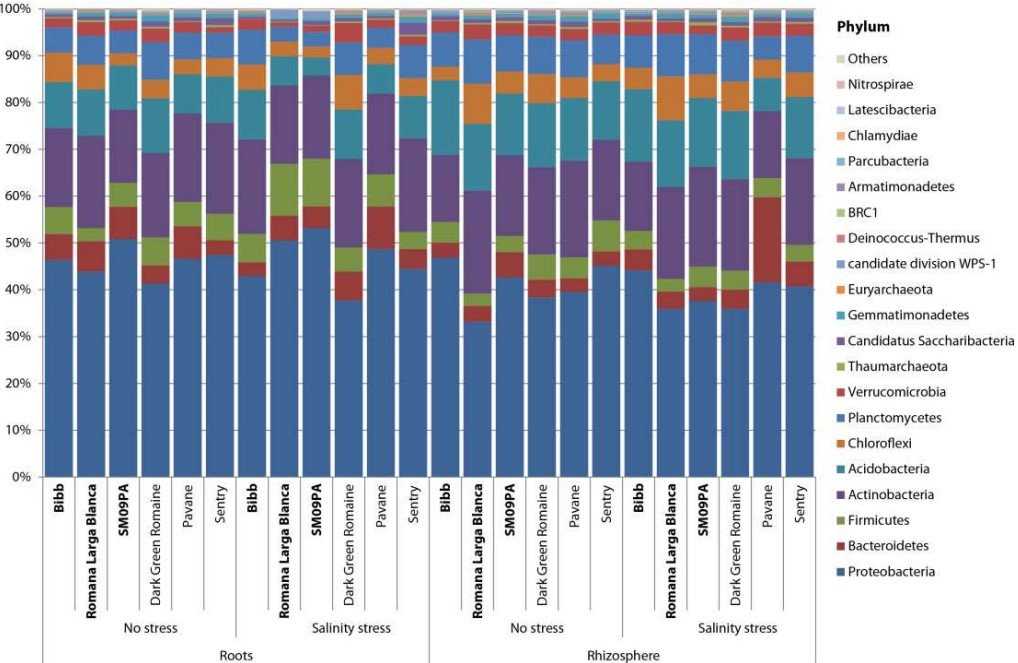

**Figure 2.** Bar chart of bacterial phyla composition in roots and rhizosphere of six lettuce varieties in normal conditions and under salinity stress.

Due to the significant interactions of factors, the microbiome response to salinity stress was analyzed separately for each sample type (Supplement Tables S4 and S5).

The most significant changes were found in the varieties Romana Larga Blanca and SM09PA, where some genera were changed by up to five two-fold logs.

In these varieties, *Herbaspirillum* was positively affected by salinity whereas the occurrence of *Pseudomonas*, *Minicystis*, or *Enterobacter* was lower. Among the less prevalent genera, *Kineosporia*, *Mogibacterium*, *Cupriavidus*, *Methyloceanibacter*, *Pseudobacteroides*, and *Gelria* were positively affected by salinity in the roots. In rhizosphere samples, salinity affected the microbiomes of Pavane and SM09PA varieties. Microbiome reactions to salinity were often opposite in those varieties. *Pseudomonas*, *Flavobacterium*, *Cellvibrio*, and *Luteolibacter* increased in Pavane but decreased in SM09PA. There was also a very significant increase of known halotolerant bacteria *Jeotgalicoccus* found in some samples. Other halo-

tolerant genera such as *Halomonas*, *Halobacillus*, or *Thiohalobacter* showed an increase, but also decreased depending on the variety and sample type.

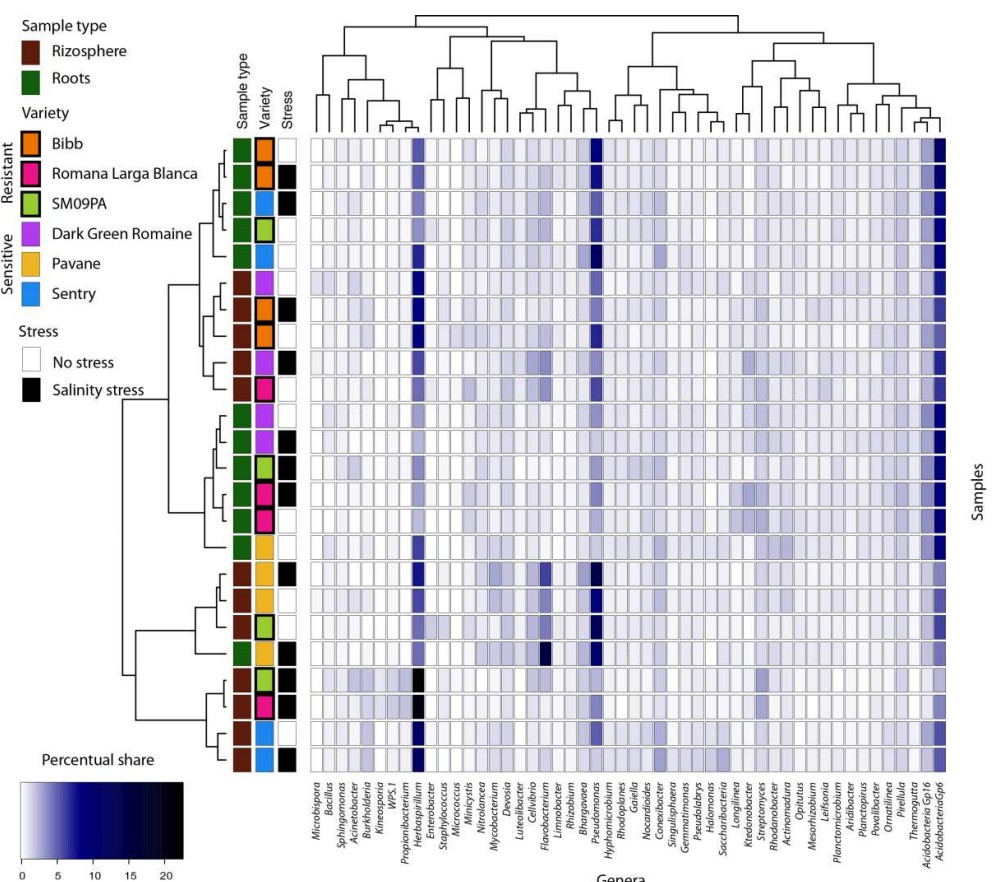

**Figure 3.** Heatmap of bacterial genera composition in roots and rhizosphere of six lettuce varieties in normal conditions and under salinity stress. Only genera with min. 2% occurrences in any samples are listed. Dendrograms are based on occurrence of genera in samples (Bray–Curtis distance, complete clustering).

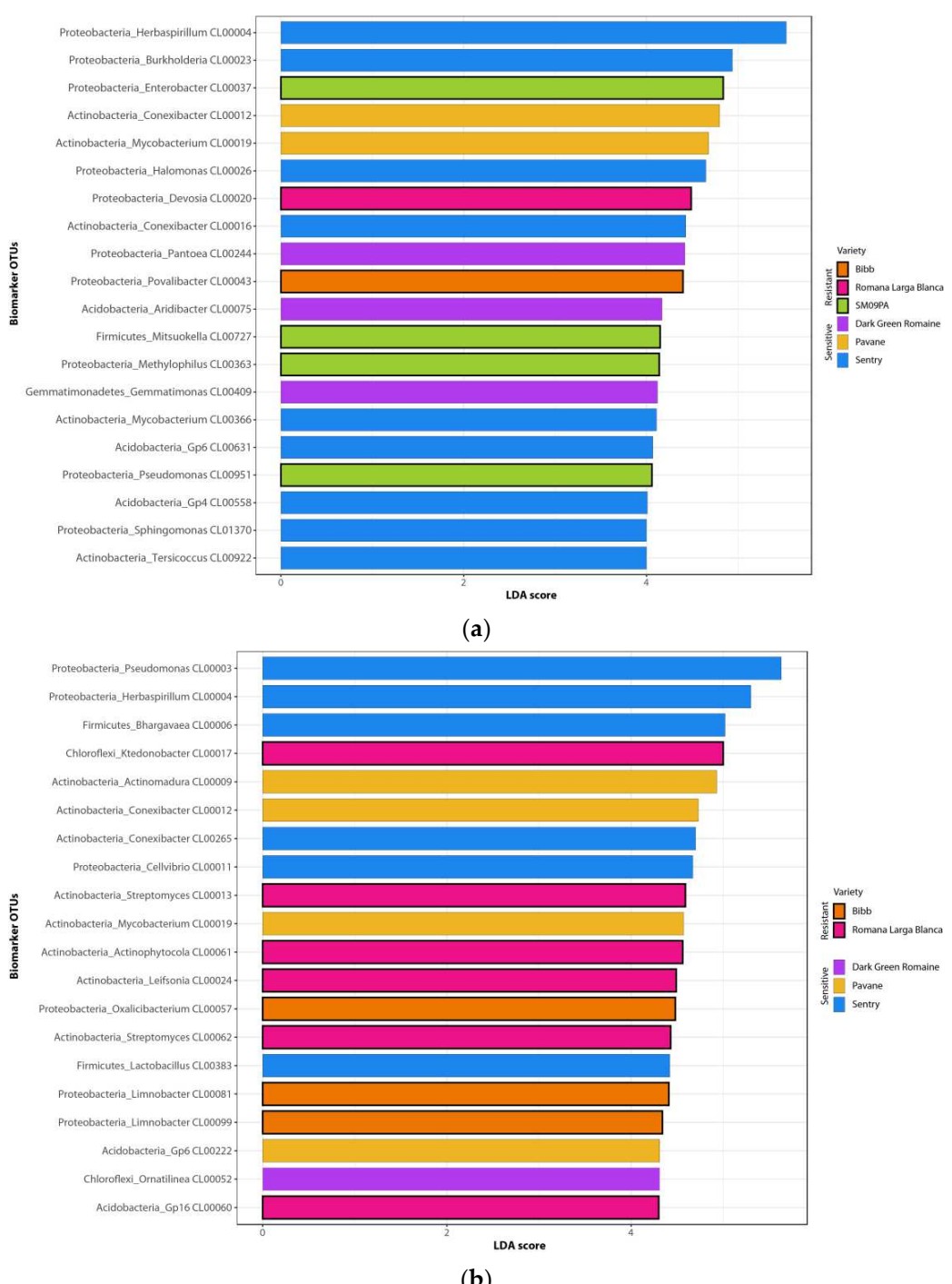

**Figure 4.** Bar charts of the first 20 LefSe biomarkers among bacterial OTUs in roots (**a**) and rhizosphere (**b**) of six lettuce varieties.

### 3.2. Diversity and Structure of Fungal Community

Similar to the bacteria, most of the difference between the diversity estimates of fungi (Table 2) was attributed to sample type, where all indices were significantly lower in the root samples than in the rhizospheres ($p < 0.001$ for all indices). The effect of the variety was lower ($p < 0.001$, $p = 0.036$, and $p = 0.029$ for richness, evenness, and Shannon's index respectively). Contrary to bacterial diversity, salinity stress was not a significant factor for any of the indices ($p = 0.565$, $p = 0.890$, and $p = 0.984$). However, certain interactions of stress and variety were found ($p = 0.002$, $p = 0.003$, and $p < 0.001$) and there was great variance among samples (Supplement Table S6).

**Table 2.** Alpha diversity indices of fungal microbiomes in root and rhizosphere of six varieties of lettuce under normal conditions and salinity stress.

| Factor | Variant | Richness | Shannon's Index | Evenness |
|---|---|---|---|---|
| Sample type | Root | 114 a * | 3.64 a | 0.510 a |
| | Rhizosphere | 158 b | 4.68 b | 0.611 b |
| | | $p < 0.001$ | $p < 0.001$ | $p < 0.001$ |
| Variety | Bibb | 175 b | 4.51 b | 0.566 ab |
| | Dark Green Romaine | 113 a | 3.99 ab | 0.563 ab |
| | Romana Larga Blanca | 128 a | 3.82 a | 0.512 a |
| | Pavane | 118 a | 4.04 ab | 0.568 ab |
| | Sentry | 140 ab | 4.18 ab | 0.559 ab |
| | SM09PA | 141 ab | 4.4 ab | 0.594 b |
| | | $p < 0.001$ | $p = 0.029$ | $p = 0.036$ |
| Salinity stress | No | 134 a | 4.15 a | 0.560 a |
| | Yes | 138 a | 4.17 a | 0.561 a |
| | | $p = 0.565$ | $p = 0.890$ | $p = 0.956$ |

* Averages followed by the same letter are not significantly different within each factor (ANOVA, Tukey test $\alpha = 0.05$).

The composition of the fungal microbiome was quite different between the sample types (PERMANOVA $p < 0.001$; $R^2 = 0.25$). In the NMDS plot (not shown), samples created distinct groups and again, their dispersions were significantly different (BETADISPER $p < 0.001$). In the rhizosphere, lettuce varieties formed clearly separated clusters (PERMANOVA $p < 0.001$; $R^2 = 0.48$) but any grouping of root samples according to the variety in the NMDS plot was not obvious (PERMANOVA $p = 0.086$) (Figure 5). The pairwise comparison between the varieties demonstrated significant results in the rhizosphere only (Supplement Table S1). Salinity stress did not cause significant changes of fungal microbiome in the root samples although differences caused by salinity stress in the rhizosphere were significant (PERMANOVA $p < 0.001$; $R^2 = 0.15$). Interactions between the variety and salinity stress (PERMANOVA $p < 0.001$) indicated different reactions of the root and the rhizosphere microbiomes to stress in particular varieties.

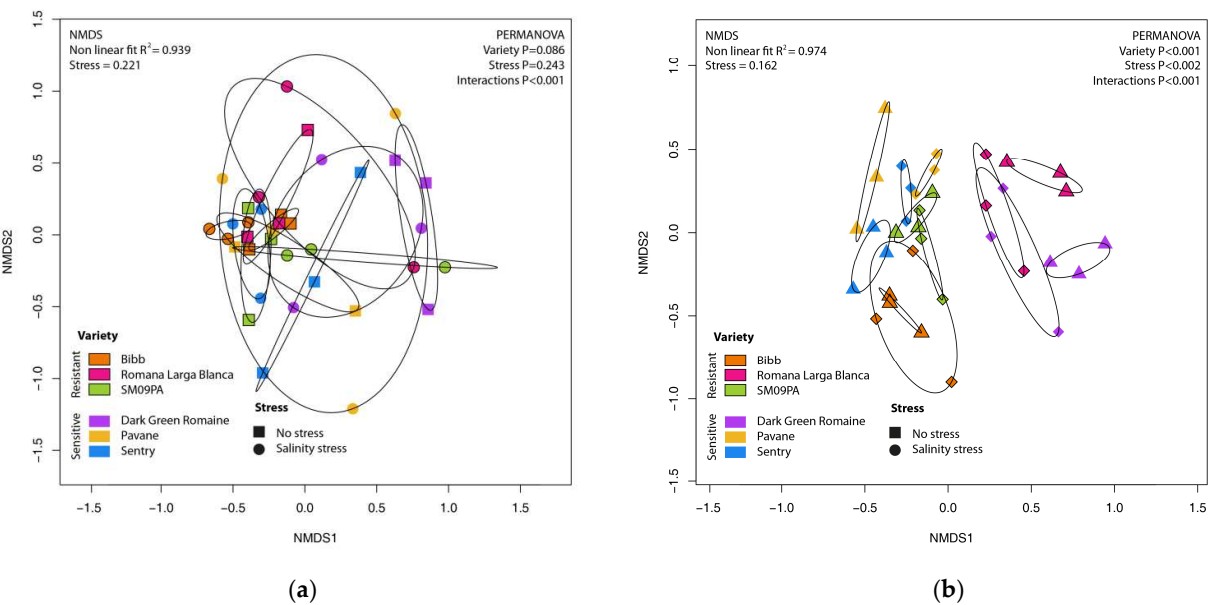

(**a**)　　　　　　　　　　　　　　　　　　(**b**)

**Figure 5.** NMDS scatterplot of fungal community in roots (**a**) and rhizosphere (**b**) of six lettuce varieties in normal conditions and under salinity stress.

Significantly less *Zygomycota* but more *Basidiomycota* were observed in the root samples (Figure 6). Furthermore, most prevalent genera were different in the sample types (Figure 7). *Actinomucor* and *Arthrobotrys* were the most prevalent in the roots whereas *Candida* and *Malassezia* were the most prevalent in the rhizosphere.

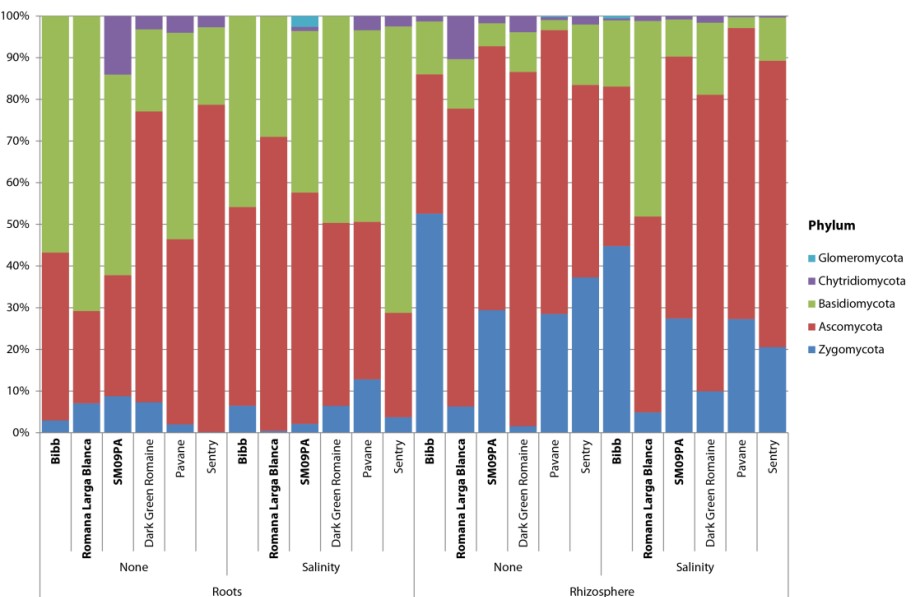

**Figure 6.** Bar chart of fungal phyla composition in roots and rhizosphere of six lettuce varieties in normal conditions and under salinity stress.

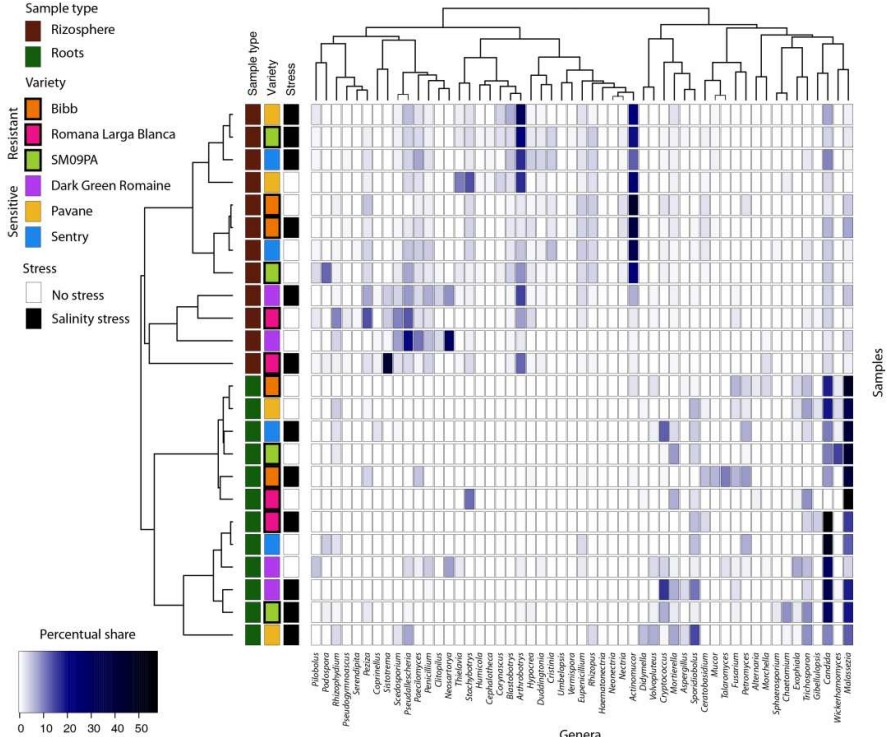

**Figure 7.** Heatmap of the most common fungal genera in samples from lettuce roots and rhizosphere. Only genera with min. 3% occurrences in any samples are listed. Dendrograms are based on occurrence of genera in samples (Bray–Curtis distance, complete clustering).

*Actinomucor*, *Malassezia*, and *Rhizopus* were identified as biomarker taxa for the Bibb variety whereas *Sistotrema* and *Scedosporium* were identified for the Romana Larga Blanca

variety. *Arthrobortys*, *Thielavia*, and *Blastobotrys* were biomarkers for the Pavane variety. *Neosartorya*, *Pseudaellescherichia*, *Paecilomyces*, and *Penicillium* were found for the Dark green variety, and *Cristinia* for the Sentry variety. The biomarkers for SM09PA were *Podosphora* and *Pilobolus*. Despite different biomarker genera, some OTUs from the same fungal species were found as biomarkers for certain varieties (Figure 8). LefSe analysis did not show any statistically significant results for any variety in the root samples.

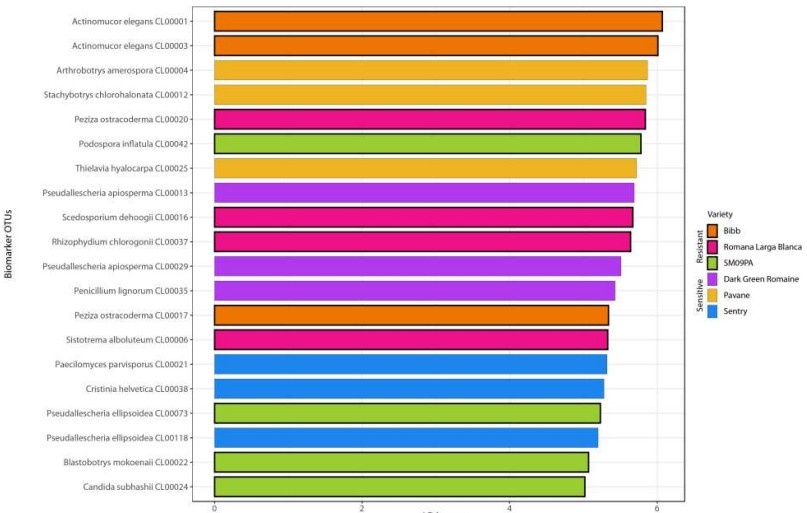

**Figure 8.** Bar chart of the first 20 LefSe biomarkers among fungal OTUs in rhizosphere of six lettuce varieties.

In the root samples, there was no differently abundant taxon in response to salinity stress. However, several differentially abundant taxa were found for samples of the rhizosphere (Supplement Table S7). Among them, *Vermispora* was the only one that was solely positively affected, whereas others increased and decreased in certain varieties.

## 4. Discussion

Plant microbiomes are currently considered an integral part of the plant holobiont system, where the synergy between plants and related microbiota provides important system services [57,58]. However, plant pathogenic or human pathogenic members of microbiome can cause loss of yields or possess a threat for consumer health [59]. This study examined microbiome changes in the roots and rhizosphere of six varieties of lettuce. One of the main characteristics of varieties was their resistance to salinity, thus microbiomes were also examined in salinity stress conditions, and both factors affected the microbial diversity and community.

The microbiome of lettuce has been targeted by several studies that analyzed the impact of soil conditions and agricultural practices on the composition of microbiomes. In the study by Schreite, et al. [60], the bacterial community in the rhizosphere of field-grown lettuce was analyzed by denaturing gradient gel electrophoresis and pyrosequencing of the 16S rRNA gene. The microbiome was mainly shaped by the type of soil and the stage of development. In all three studied soil types, *Proteobacteria* phyla was enhanced in lettuce rhizosphere. *Sphingomonas*, *Rhizobium*, and *Pseudomonas* were among the most abundant genera. In the study of Iliev et al. [61], 98% of sequences came from nine phyla (Proteobacteria, Actinobacteria, Firmicutes, Acidobacteria, Chloroflexi, Bacteroidetes, Gemmatimonadetes, Verrucomicrobia, and Nitrospira). Cardinale et al. [62] reported that *Proteobacteria*, *Bacteriodetes*, *Chloroflexi*, and *Actinobacteria* dominated in the lettuce root microbiome. The results from those authors are in concordance with the findings of this study, where the most common phyla were *Proteobacteria*, *Actinobacteria*, *Acidobacteria*, *Planctomyces*, *Firmicutes*, *Chloroflexi*, and *Bacteroidetes*.

Iliev et al. [61] also showed that conventional fertilization reduced the diversity of bacteria in lettuce-related rhizospheres and recommended bio-organic fertilizers because they can increase the occurrence of bacteria previously known to suppress plant pathogens. Fertilization of lettuce by feather-based compost significantly changed the composition of the rhizosphere microbiome. [63].

Sun et al. [64] analyzed the effect of manure fertilization of lettuce. They also showed a significant shift in the bacterial community and altered resistome in the soil end episphere of lettuce. However, the endosphere of lettuce remains almost unchanged, which is important in terms of consumer safety. Erlacher et al. [65] pointed out some potentially human pathogenic bacteria within the rhizosphere and phyllosphere of lettuce. Later, the effect of biotic stress on the abundance and structure of *Enterobacteriaceae* was analyzed [66].

*Enterobacteriaceae* and other potential human pathogenic bacteria in lettuce microbiomes has gained great attention as lettuce is consumed in its fresh state, and any contamination can result in health issues. Several microbiome studies assessed the microbial community structure in lettuce leaves to examine potential human pathogens. Yeon-Cheol, Su-Jin, and Da-Young [23] used 16S rRNA gene-based sequencing to identify foodborne pathogens in lettuce during different seasons and the potentially pathogenic bacteria such as *Bacillus* spp., *Enterococcus casseliflavus*, *Klebsiella pneumonia*, and *Pseudomonas aeruginosa* were identified. Within this study, the presence of the genus *Enterobacter* achieved up to 1.5% of the total community in the roots of the SM09PA variety. It was also detected in the roots of other varieties as well. Furthermore, members of the bacterial genera *Pseudomonas*, *Pantoea*, or *Burkholderia* that were very common in the root samples can be potentially pathogenic in specific cases [22]. On the other hand, certain strains of *Pantoea*, *Pseudomonas*, or *Acinetobacter* may provide some growth promotion and disease resistance for plants [67]. Some human pathogenic bacteria live internalized in plant tissue and transmission between the internal tissue of the root and the consumed parts of the plant (leaves) is presumed. Metagenomic sequencing of lettuce leaves indicated that the pre-storage bacterial community is variable, usually dominated by the species *Erwiniaceae* and *Pseudomonadaceace*, and after cold storage, differences based on varieties emerge [68].

The association of microbiomes to the plant genotype is less commonly studied, even in other plant species. The performed studies of lettuce microbiota showed that the structure of the phyllosphere microbiota is more influenced by the morphological difference of lettuce phenotypes than the lettuce genotype [69,70]. However, the root-associated microbiota needs to be studied and defined more precisely. Comparison of root microbiota in ancient and modern lettuce varieties and its wild ancestor *Lactuca serriola* showed that the domestication of lettuce led to the diversification of bacteria in the root system [62]. This study shows that different microbiomes could develop in the root area of certain lettuce varieties. Significantly different diversity and community composition were found among varieties. *Herbaspirillum, Enterobacter, Burkholderia, Conexibacter, Mycobacterium*, and other groups of bacteria were more abundant in certain varieties. Specific microbiome development may be caused by physiological properties of the varieties, their pathogen/microbe resistance system, growth ability, accumulation of nutrients, organic matter production, and rhizodeposition [15]. Due to these factors, only a portion of microbial species/strains can colonize root tissue and potentially other parts of plants. Root exudates play a key role in microbial development in the root zone [71]. Exudates are the main negotiation mechanism between plants and microorganisms in the surrounding soil [72]. Plant genotypes with a certain composition of exudates have specific microbiomes in the rhizosphere [73]. Modern varieties with faster and stronger development of roots usually harbor more diverse microbial communities [74]. Reciprocally, microorganisms in the rhizosphere can promote plant growth by producing molecules that modulate the growth of plants such as fytohormones. Jasmonic acid, salicylic acid, ethylene, cytokinins, gibberellic acid, abscisic acid, auxins, and others are produced by various species of bacteria [75]. Microorganisms can also increase the availability of nutrients (e.g., phosphorus) and thus increase plant growth [76].

As the results show, a large part of the bacterial community was shared between the rhizosphere and root internal tissue despite significantly fewer species/OTUs within the roots. On the other hand, the fungal community contained distinct species/OTUs for the root tissue and the rhizosphere. The internalizing ability of fungi is generally lower than bacteria [77]. Yeast or yeast-like species were most common in root tissues with great variability between samples. Fungal communities in the rhizosphere were shaped similarly to bacterial ones, probably due to the root exudates. Microbial loads on the surface of seeds or bacteria internalized in the seed can directly affect the microbiome of developing plants, including the rhizosphere [78]. These microbes are naturally selected and already adapted to certain conditions of plant/variety. Despite surface sterilized seeds in this assay, it is still necessary to consider the possibility that distinct development of microbiomes across varieties was caused by their primary microbial load as the microorganisms are internalized within the seed [79].

Both biotic and abiotic stresses are usually associated with microbiome changes [80]. In this study, salinity stress caused significant changes in the community. However, the effect was dependent on variety. The effect of salinity on the root associated microbiome was significant only in the bacterial community. The fungal community did not respond to the salinity stress.

Many authors [33,81–83] found negative correlations between soil salinity and bacterial diversity. It is based on the elevated extracellular osmolarity that leads to damage of membranes as well as proteins and nucleic acids of the bacteria. The microbial diversity decreases as a consequence, because only some species are able to adapt to these conditions. The results of this study confirmed this hypothesis. Liu et al. [84] found *Proteobacteria* to be less sensitive to salinity as their frequency rose alongside a salinity gradient whereas *Actinobacteria* decreased.

Specific microbiome reactions were found mainly in varieties that were more resistant to salinity. In salinity conditions, sensitive varieties are stressed and would greatly change their growth. Comparing the habitus, plants of the sensitive varieties were smaller than resistant varieties in the assay and probably shortened their production of rhizodeposits. Resistant varieties with maintained growth still provided a high amount of metabolites resulting in the development of specific microbiomes. Fungi are generally less sensitive to osmotic stress, and they also grow significantly slower than bacteria [85]. The combination of such factors with relatively low levels of osmotic stress and short examined periods resulted in insignificant changes. Specific microbiomes develop on the roots of plants growing in saline environments [34]. From another viewpoint, specifically developed microbiomes can help plants survive in saline conditions [36]. Modification of microbiomes may be a viable way to enhance plant tolerance to salinity stress [86,87]. For example, inoculation of rice seeds with halotolerant microbiomes obtained from marine sediment or rice fields led to the improved growth of the rice plants under salinity stress [88]. According the results, plant variety must be considered when such an inoculation is applied in the field.

## 5. Conclusions

The results of this study showed that the rhizosphere and root internal microbiomes are significantly affected by lettuce variety. Moreover, the microbiome of roots reacts to osmotic stress differently in certain varieties, and it seems to be related with the variety resistance to osmotic stress. Different growth of resistant and sensitive varieties is likely to be the basis of microbiome changes in salinity conditions. Changes in plant microbiomes may have consequences for plant health, yield amount and quality, and consumer safety. However, further studies are needed to determine the reasons and outcomes of the specific changes in lettuce microbiomes under osmotic stress.

**Supplementary Materials:** The following supporting information can be downloaded at: https://www.mdpi.com/article/10.3390/horticulturae8121174/s1, Table S1. Primers used for amplification

of 16S rRNA gene and ITS2 region. Table S2. One-way ANOVA comparison of alpha diversity indices of bacterial microbiome in root and rhizosphere of six varieties of lettuce under normal conditions and salinity stress. Table S3. Pairwise comparison of root and rhizosphere microbiome between six varieties of lettuce. Table S4. Changes of genera abundance in bacterial community in roots of six varieties of lettuce under salinity stress. Only genera significantly changed in at least single variety are listed. Values are two-fold logs, ** $p < 0.01$, * $p < 0.05$. Table S5. Changes of genera abundance in bacterial community in rhizosphere of six varieties of lettuce under salinity stress. Only genera significantly changed in at least single variety are listed. Values are two-fold logs, ** $p < 0.01$, * $p < 0.05$. Table S6. One-way ANOVA comparison of alpha diversity indices of fungal microbiome in root and rhizosphere of six varieties of lettuce under normal conditions and salinity stress. Table S7. Changes of genera abundance in fungal community in rhizosphere of six varieties of lettuce under salinity stress. Only genera significantly changed in at least single variety are listed. Values are two-fold logs, ** $p < 0.01$, * $p < 0.05$.

**Author Contributions:** Conceptualization, J.Ž. and J.M.; methodology J.M.; software J.M.; validation, J.Ž., L.U. and R.O.; formal analysis, L.U., R.A. and J.M.; investigation, J.Ž., L.U., R.A., R.O. and J.M.; resources R.O. and J.M. data curation J.M.; writing—original draft preparation, J.Ž., L.U. and J.M. writing—review and editing, J.Ž. and D.M.; project administration J.M. All authors have read and agreed to the published version of the manuscript.

**Funding:** This research was financially supported by a project of the Ministry of Education, Science, Research and Sport of the Slovak Republic, grant no. VEGA 1/0661/19 "Plant microbiome and safe food".

**Institutional Review Board Statement:** Not applicable.

**Informed Consent Statement:** Not applicable.

**Data Availability Statement:** All data are available from the correspondence author upon request. Sequence data were deposited in GenBank databases under BioProject accession No. PRJNA893639.

**Acknowledgments:** The authors would like to thank to Ivan Šimko, USDA Agricultural Research Service, U.S. Department of Agriculture, for providing the biological material and Katarína Ražná for the set-up of the assay-growing conditions.

**Conflicts of Interest:** The authors declare no conflict of interest.

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
