# Peer review of "Varieties of Lettuce Forming Distinct Microbial Communities Inhabiting Roots and Rhizospheres with Various Responses to Osmotic Stress"

_horticulturae, doi:10.3390/horticulturae8121174_

Round 1

Reviewer 1 Report

These study covers quite new area. The results and the way of interpretation and presentation of the experiment is interesting  and useful for further studies in this field.

Do you expect a similar situation with other crops? Also how these results could be helpful in the production of samples. Could they be used to help plant growth and pathogen defence systems? General comment some more practical benefits of the obtained results please add.

This kind of studies are future so I recommend this paper for publishing.

Reviewer 2 Report

Dear authors, I have read with interest the manuscript entitled "Varieties of lettuce forming distinct microbial communities inhabiting roots and rhizosphere with various responses to osmotic stress". The idea is very important for future researches that deals with plant rhizosphere microbiome as affected and shaped by different stresses.

There are some suggestions that I consider will improve your work.

Avoid more than 5 lines long sentences. Try to split them in two different sentences. Also, rewrite the sentences where you use a personal addressing (e.g. "our") into a more impersonal form.

Specific comments - line 100 - personal communication - add a site or a reference instead of this source.

In table 1 and 2 you present single factor interaction. It will be interesting to present all three factors interaction (Sample type * Variety * Salinity stress). This will present better your results and also will permit you to make a hierarchy of each combination and to identify the highest values for each variety (as an example). You can introduce this interaction as a supplementary file.

Overall, the manuscript is interesting and well constructed.

Reviewer 3 Report

1. It is known that in field and greenhouse conditions lettuce is not irrigated with distilled water. Do you think the rhizosphere microbiome will be very different compared to your experiment?

2. Why exactly this concentration of NaCl was taken in the experimental group? Did salinity affect plant habitus?

3. I think it is necessary to increase the number of replications in the following experiments.

3. It is necessary to expand the discussion by adding the proposed specific mechanisms of the effect of salt stress on the  rhizosphere microbiome.

4. Check superscripts R2, for example at lines 192, 267, 270, 275.

5. Check capital letter at line 279 Figure 7.

Reviewer 4 Report

Žiarovská and colleagues present a comparative study on root and rhizospheric bacterial and fungal communities in six different lettuce varities exposed to salinity stress.  They authors state correctly in the introduction that understanding the biological processes shaping microbial structures is an important task to ensure plant productivity. However, the presented study is purely descriptive and includes no mechanistic aspects. The authors should elaborate a bit more in the introduction what is new about their study since it is known that lettuce genotypes shape the microbiome composition.

 For Figure 2 and 6, I would suggest grouping the varieties according their salt tolerance.

 "For analysis of 134 fungal community, primers ITS7 and ITS4 [45] were used for amplification of ITS2 region." This source describes two different ITS7 primers, which one was used by the authors? Maybe add a supplementary table with the primers that were used in the study.

 The manucript should be revised carefully for typos, e.g. line 141.

Please also revise the reference section according to the journals requiremets.

Round 2

Reviewer 2 Report

Dear authors, the new form of the manuscript presents very well your research and results.

Reviewer 3 Report

Authors improved the manuscript. I do not have any comments or questions.

Reviewer 4 Report

One remark: Please check upper and lower cases in the references and fix them.